# Effects of UV-B and UV-C Spectrum Supplementation on the Antioxidant Properties and Photosynthetic Activity of Lettuce Cultivars

**DOI:** 10.3390/ijms25179298

**Published:** 2024-08-27

**Authors:** Ernest Skowron, Magdalena Trojak, Ilona Pacak

**Affiliations:** 1Department of Environmental Biology, Jan Kochanowski University of Kielce, Uniwersytecka 7, 25-406 Kielce, Poland; magdalena.trojak@ujk.edu.pl; 2Institute of Chemistry, Jan Kochanowski University of Kielce, Uniwersytecka 7, 25-406 Kielce, Poland; ilona.pacak@gmail.com

**Keywords:** lettuce, indoor farming, spectrum supplementation, ultraviolet light, antioxidant potential, functional foods, nutritional quality, phytochemicals, sensory properties

## Abstract

Indoor farming systems enable plant production in precisely controlled environments. However, implementing stable growth conditions and the absence of stress stimulants can weaken plants’ defense responses and limit the accumulation of bioactive, health-beneficial phytochemicals. A potential solution is the controlled application of stressors, such as supplemental ultraviolet (UV) light. To this end, we analyzed the efficiency of short-term pre-harvest supplementation of the red–green–blue (RGB, LED) spectrum with ultraviolet B (UV-B) or C (UV-C) light to boost phytochemical synthesis. Additionally, given the biological harm of UV radiation due to high-energy photons, we monitored plants’ photosynthetic activity during treatment and their morphology as well as sensory attributes after the treatment. Our analyses showed that UV-B radiation did not negatively impact photosynthetic activity while significantly increasing the overall antioxidant potential of lettuce through enhanced levels of secondary metabolites (total phenolics, flavonoids, anthocyanins), carotenoids, and ascorbic acid. On the contrary, UV-C radiation-induced anthocyanin accumulation in the green leaf cultivar significantly harmed the photosynthetic apparatus and limited plant growth. Taken together, we showed that short-term UV-B light supplementation is an efficient method for lettuce biofortification with healthy phytochemicals, while UV-C treatment is not recommended due to the negative impact on the quality (morphology, sensory properties) of the obtained leafy products. These results are crucial for understanding the potential of UV light supplementation for producing functional plants.

## 1. Introduction

Traditional agriculture faces challenges, including climate change, extreme weather conditions, land degradation, dwindling freshwater supplies, and urbanization [1]. These factors complicate the task of securely providing high-quality food for a growing population. Consequently, there is increasing interest in plant production within closed facilities, such as plant factories, vertical farms, and indoor-growing modules [2]. Indoor farming (IF) involves cultivating plants inside buildings, often without soil, using nutrient solutions and artificial lighting, allowing for year-round growth [3], and mitigating the disadvantages of open-field farming, such as weather extremes, pathogens, and pests [1]. Additionally, for most plant production, artificial lighting systems with non-saturating light intensity are adequate, with light quality being more critical than quantity [4]. 

Plants grown in open-field conditions are exposed to sunlight, which includes UV radiation [5]. Consequently, they have evolved various metabolic and biochemical responses to UV exposure, including increased synthesis of secondary metabolites (SMs) [6] and other antioxidants [7]. SMs accumulate mostly in epidermal plant layers and function as sunscreens, protecting underlying tissue from the damaging effects of UV light. Still, prolonged exposure to UV light might lower their protective potential and reduce overall photosynthetic activity [8]. Previous research [9] showed that greenhouse-grown lettuce was rather poor in total phenolic concentration and antioxidant capacity compared to plants transplanted to open field conditions, as a result of reduced overall light intensity and UV depletion in the spectrum caused by polyethylene film covering the tunnels, which absorbs shorter UV wavelengths. Still, however, for the open-field-grown lettuce, reduced biomass accumulation was noted due to inadequate soil water content, too high light intensity, and strong wind [9]. Thus, reasonable solutions in the face of climate change are attempts to adapt indoor farming for sustainable leafy vegetable cultivation. Based on the previous research [10], the most economically justified method is to grow plants continuously under UV-free conditions and then, shortly before the final harvest, expose them to UV light to achieve both high yields of biomass and increased phytochemical content complementarily. 

The UV spectrum (100–400 nm) is divided into three sub-regions: UV-A (315–400 nm), UV-B (280–315 nm), and UV-C (100–280 nm) [11]. The underlying mechanism of plant responses to UV light involves various photoreceptors, such as blue light/UV-A photoreceptors like cryptochromes (CRYS) and the UV Resistance Locus 8 (UVR8) photoreceptor, which operates through UV-B light [12]. The perception and response to UV-C light are linked to the redox state of cells and the generation of reactive oxygen species (ROS) [13]. The primary regulator of plants’ responses to UV light is the elongated hypocotyl 5 (HY5) transcription factor, whose UV-dependent accumulation induces the biosynthesis of SM such as phenolic compounds, with flavonoids being the largest class [13,14,15,16]. In the case of antioxidants, UV radiation also upregulates the expression and activity of enzymes associated with ascorbic acid (AsA) recycling [7]. Consequently, UV radiation is considered a tool to biofortify indoor farming (IF)-grown crops with functional phytochemicals and boost the antioxidant capacity of leafy vegetables [17,18]. Among the analyzed phytochemicals, polyphenols have anti-mutagenic, chemopreventive, and anti-carcinogenic activities [19], act as natural reactive oxygen species reducers, and are potent antioxidants that prevent oxidative damage to biomolecules. The main classes of polyphenolic compounds found in plants are flavonoids, phenolic acids, lignans, and stilbenes [20]. Flavonoids are the most abundant polyphenolic compounds in food and are classified as chalcones, flavanones, flavonols, flavones, isoflavones, 3-deoxyflavonoids, proanthocyanidins, and anthocyanins—the glycosylated anthocyanidins [15]. 

Therefore, this study aimed to examine the efficiency of short-term supplementation of the spectrum with UV-B (311 nm) and UV-C (254 nm) in increasing the antioxidant potential of leafy plants, as assessed by the total phenolic (TPC) and flavonoid (TFC) contents as well as the anthocyanin and ascorbic acid levels. For this purpose, *baby leaf* lettuce (*Lactuca sativa* var. *crispa* L.) cultivars with green (cv. Lollo Bionda) and reddish leaves (cv. Lollo Rossa) were grown in a growth chamber under a red-green-blue (RGB) spectrum, supplemented with increasing doses of UV light shortly prior to harvest. Then, to indicate the influence of UV light exposition, we monitored photosynthetic activity with chlorophyll *a* fluorescence as well as the level of photosynthetic pigments, proteins, and lipid peroxidation rate. While assessing the efficiency of UV treatment on the antioxidant potential, we also carried out analyses of the overall antioxidant potential (TAC, total antioxidant capacity) and showed that UV-B exposure enhanced TPC, TFC, and anthocyanins in both lettuce cultivars, while ascorbic acid was only present in green leaf one. At the same time, plants treated with a UV-B cumulative dose (CD) of 15.622 kJ m^−2^ show no negative impact on the photosynthetic apparatus functionally of the plants’ morphology. On the other hand, despite the restricted exposition time to UV-C light (CD = 6.008 kJ m^−2^), its application significantly decreases the photochemical activity and reduces the rate of controlled energy quenching, especially in green leaf lettuce. Moreover, UV-C treatment has been significantly less effective to induce SM and ascorbic acid accumulation than UV-B. Also, the UV-C-treated lettuce of both cultivars presented undesirable morphological traits such as leaf bronzing and curling and consequently lower scores within consumers’ sensory attributes tests. This study provides valuable insights into the role of UV-B and UV-C supplementation in standard RGB lighting systems, which are mostly devoid of UV components, to improve the quality of leafy plant products.

## 2. Results

### 2.1. Antioxidant Capacity in Response to Supplemental UV-B or UV-C Light

#### 2.1.1. Total Phenolic Content

The estimated total phenolic content (TPC) is expressed as µg gallic acid equivalents per mg of fresh weight (FW) (Figure 1). Analysis showed that a reddish cultivar of lettuce presented almost 3 times higher phenolic content under the control RGB spectrum (Figure 1b) than the green-leaf one (Figure 1a). The short-term exposition of plants to supplemental UV-B light increased TPC levels, especially in green-leaf cultivars, as we observed 28% and 7% higher TPC levels compared to control for LB and LR, respectively. In contrast, UV-C exposure did not stimulate TPC accumulation in the LB cultivar and even decreased its level by approximately 18% in the LR one.

#### 2.1.2. Total Flavonoid Content

Total flavonoid content (TFC) is expressed as µg rutin equivalents per mg of fresh weight (FW) (Figure 2). Similarly to TPC, flavonoid content was significantly higher in the reddish leaf cultivar Lollo Rossa compared to Lollo Bionda. Under the RGB spectrum, we documented a 2.7 times higher TFC level in the LR compared to the LB cultivar. Supplementation of the RGB spectrum within UV-B light increased flavonoid content by almost 3- and 1.6-fold in LB (Figure 2a) and LR cultivars (Figure 2b), respectively. As in the case of TPC, also in TFC estimation, UV-C showed a negative impact in the LR cultivar, with 17% lower TFC compared to the RGB group (Figure 2b). The Lollo Bionda cultivar showed, however, a 2-fold increase in the TFC pool after UV-C exposure (Figure 2a).

#### 2.1.3. Anthocyanins Level

Under the category of flavonoids, anthocyanins (ANT) are prominent compounds that naturally occur as glycosides in pigmented organs of plants. As expected, the reddish lettuce LR showed a 5.5 times higher ANT level compared to the green leaf LB (Figure 3). Moreover, analyses showed that UV-B supplementation to the RGB spectrum significantly increased the ANT pool in both green and reddish lettuce cultivars, as LB lettuce showed 15% (Figure 3a) and LR—71% (Figure 3b) higher ANT levels compared to RGB control after UV-B exposure. Like the TFC level, UV-C exposition increased the ANT level in LB cultivars by 46% while reducing its accumulation in LR by 30% compared to RGB plants.

#### 2.1.4. Ascorbic Acid Pool

In addition to polyphenolic SM, we also analyzed the ascorbic acid pool in both lettuce cultivars, both an initial AsA pool and total AsA level, after reduction of its residual oxidized form, the dehydroascorbic acid (DAsA), back to AsA with reducing agent—DTT. Interestingly, in the case of the reddish cultivar LR of lettuce, most of the AsA level was presented in reduced AsA form, except for the pool observed for the UV-C-supplemented plants (Figure 4b). UV-B and UV-C induced a total pool of AsA (AsA + DAsA) by 33 and 16%, respectively. At the same time, LB plants grown under the RGB spectrum showed an almost 1.5 times lower total AsA pool compared to LR plants. LB plants have also been more vulnerable to UV-B light exposure, as while the reduced form of AsA remained at an RGB-like level, plants exposed to UV-B accumulated significantly higher levels of DAsA (Figure 4a). Such observation has also been proven within the AsA + DAsA/AsA ratio estimation, which increased in the LB cultivar after UV-B exposition by 85% (Figure 4c). In the case of the reddish cultivar, no changes in the mentioned ratio have been documented (Figure 4d).

#### 2.1.5. Overall Antioxidant Capacity

The total antioxidant capacity (TAC) was expressed as µg equivalents of butylated hydroxytoluene (BHT) per mg of FW (Figure 5a,b) and DPPH radical scavenging activity rate (Figure 5c,d), which were assessed based on fitted experimental data of the BHT calibration curve, as described in Materials and Methods. As expected, analysis proved that overall antioxidant capacity complies with polyphenolic SM, especially with TPC in LB or TFC and ANT level in the LR cultivar. Under a spectrum depleted of UV light radiation, the green leaf cultivar showed 1.5 times lower antioxidant capacity compared to the reddish one. Despite the short-term exposition of plants to UV-B or UV-C light, it was efficient in increasing the antioxidant capacity of the green cultivar. In the LB lettuce, TAC was increased by 66 and 53% for UV-B and UV-C, respectively (Figure 5a). Also, radical scavenging activities were increased in UV-B and UV-C by 10 and 8%, respectively. 

On the contrary, even though LR lettuce showed enhanced polyphenolic SM level and AsA pool in response to UV-B treatment, the overall antioxidant capacity showed no further increase, presumably due to the initial high level of antioxidant compounds. Thus, the analyzed DPPH radical scavenging activity reached a control-like value of 78% (Figure 5d). At the same time, UV-C exposition exerted a significant negative influence on TAC due to reduced TPC, TFC, and ANT levels, and a consequently 10% lower scavenging activity of the applied radical.

### 2.2. Photosynthetic Activity under Short-Term Exposition to UV-B or UV-C Light

#### 2.2.1. The Effect of UV Light Supplementation on Photosynthetic Pigments and Soluble Leaf Protein Content

Both analyzed lettuce cultivars presented similar content of chlorophyll *a* + *b* under the RGB spectrum (Table 1 and Table 2). As expected, however, the increased level of screening pigments in reddish Lollo Rossa lettuce protects chlorophylls from UV-driven degradation, as noticed with almost unchanged levels in both UV treatments (Table 2). At the same time, green-leaf Lollo Bionda lettuce presented 9 and 62% decreased chlorophyll *a* + *b* content in response to UV-B and UV-C exposure, respectively. Moreover, in the case of LB plants exposured to UV-C light, a drop in chlorophyll content was noted, which was more strongly associated with the reduced chlorophyll *a* (decreased by 66%) than chlorophyll *b* (50% drop), indicating the reaction centers (RC) of photosystems were dismantled rather than antennas. Also, in the LR cultivar, UV light increased chlorophyll *b* levels by 26 and 7% for UV-B and UV-C, respectively, while the chlorophyll *a* content in the mentioned groups decreased by 10 and 8%, respectively.

The relative change in chlorophyll content has been estimated with the chlorophyll *a*/*b* ratio (Table 1 and Table 2). The accessory pigment content, measured as the carotenoid pool, also showed that UV-B light exposure increased the number of carotenoid-rich antennas per RC, as in the LR cultivar, a 15% higher level of carotenoids has been noticed compared to the RGB group. In the LB plants, no difference has been observed under UV-B supplementation. On the contrary, UV-C light caused a reduction in the carotenoids pool by 73 and 26% for LB and LR, respectively, and consequently increased the ratio of chlorophyll *a* + *b* to carotenoids in UV-C-treated plants. The total content of soluble leaf protein (SLP) has also been analyzed and showed that in the case of green leaf lettuce, both UV light expositions exerted a positive impact on SLP. UV-B and UV-C supplementation enhanced SLP in the LB cultivar by 59 and 17%, respectively. Conversely, in the LR cultivar, UV-B showed no influence on SLP, while UV-C decreased protein levels by at least one-third. Moreover, there was a noticeable lower SLP in the LB than in the LR cultivar under the RGB spectrum (Table 1 and Table 2).

#### 2.2.2. Influence of UV Light Supplementation on RuBisCO Abundance

Electrophoretic separation of SLP, followed by densitometric analysis of the RuBisCO enzyme’s large (LSU) and small (SSU) subunits within ImageJ software (v. 1.49), proved that even the short-term UV light treatment modifies RuBisCO accumulation. The relative amounts of RuBisCO LSU and SSU are essentially consistent with the SLP. In the case of the LB cultivar, UV-B treatment increased LSU and SSU by 73 and 81%, respectively, while UV-C increased LSU and SSU by 34 and 36%, respectively (Figure 6a,c). On the contrary, in LR lettuce, UV-B radiation decreased the relative abundance of LSU and SSU, compared to RGB, by 20 and 41%, respectively, while in response to UV-C, LSU and SSU dropped by 69 and 83%, respectively (Figure 6b,c).

#### 2.2.3. The Effect of UV Light Supplementation on Subsequent Photosynthetic Efficiency of PSII

To estimate the influence of the progressively increased time of UV light exposure on lettuce, we analyzed the actual condition of the photosynthetic apparatus every other day after the treatment with chlorophyll *a* fluorescence induction kinetics. Results are depicted in Figure 7 and Figure 8. In the case of UV-B light, no negative impact on the maximum quantum efficiency of the photosystem (Fv/Fm) has been noted for both cultivars even after the last day of treatment with a cumulative dose of 15.622 kJ m^−2^ (Figure 7a and Figure 8a). Although the detailed analyses of the photochemical (Figure 7b and Figure 8b) and non-photochemical energy distribution (Figure 7c,d and Figure 8c,d) at the first day (Day 1) showed a slight decrease in effective quantum yield of PSII photochemistry (ΦPSII) (Figure 7b and Figure 8b), followed by a simultaneous increase of ΦNPQ (Figure 7c and Figure 8c), the changes seem be temporal. After the fourth day of UV-B treatment, photochemical and non-photochemical energy distribution, as well as the ETR (Figure 7f and Figure 8f), regained the control level in both cultivars, indicating that plants acquired a sort of acclimation. 

On the contrary, UV-C light supplementation, despite the reduced time of exposition and CD compared to UV-B, exerted a negative impact on the photosynthetic activity of both cultivars since the first day of exposition. The harmful effects of UV-C have been noticed with reduced photochemical quenching (Fv/Fm, ΦPSII and ETR) (Figure 7a,b,f and Figure 8a,b,f) and activation of the mechanism of energy dissipation in the form of heat with the NPQ mechanism (ΦNPQ and NPQ, Figure 7c,e and Figure 8c,e), followed by increased passive energy losses documented with ΦNO. It should be noted, however, that the LR cultivar tended to acquire a sort of acclimation after the third day of UV-C treatment, with increased photochemical efficiency (Figure 8a,b) and electron transport rate (Figure 8f) and reduced yield of non-photochemical quenching (Figure 8c,d). Yet, in both cultivars, the NPQ parameter (Figure 7e and Figure 8e) remained decreased after the fourth day, which indicates that the effective protective mechanism of excessive absorbed energy dissipation in the form of harmless heat has been downregulated.

#### 2.2.4. The Effect of UV Light Supplementation on Lipid Peroxidation Rate 

Due to the noticeable negative influence of UV-C light treatment on photosynthetic apparatus status, we also analyzed the rate of oxidative stress within the TBARS assay, which detected byproducts of lipid peroxidation in the sample, mostly MDA. Surprisingly, the increased rate of TBARS formation due to increased ROS formation has been identified only in green leaf cultivars (Figure 9a), but still, the increase was limited to 24% in response to UV-B exposure, while in UV-C no additional TBARS formation has been noticed. In the case of the reddish cultivar, UV-B as well as the UV-C treatment decreased TBARS levels by 15% for both compared to RGB (Figure 9b). It should also be noted that the LR cultivar showed 2.5 times higher TBARS levels even under the RGB control spectrum compared to the LB one.

#### 2.2.5. The Effect of UV Light Supplementation on Plant Morphology and Sensory Properties

The negative impact of even a low dose of UV-C exerted on lettuce plants has been proven when analyzing the plants’ morphology of LB (Figure 10) and LR (Figure 11) cultivars, which present severely inhibited growth and misshaped, visibly damaged leaves with necrotic lesions placed on the edges of the leaf blade. Consequently, we also analyzed the way the UV light supplementation influenced the main sensory attributes of leafy vegetables relevant to the consumer: appearance, crispness, taste (sweetness, bitterness), and overall assessment within consumers, scoring blinded samples at a scale of 0–5, where 5 is the maximal and 0—the least score (Figure 12a,b).

Moreover, according to the consumers’ test, supplementation of the RGB spectrum within UV-B light reduced the sweetness and increased the bitterness of LB lettuce leaf from 3.2 (±0.63) and 3.0 (±0.47) to 2.8 (±0.42) and 3.6 (±0.52), respectively (Figure 12a). In the case of LR lettuce, sensory tests showed that consumers scored the control RGB samples as less sweet and more bitter than LB (Figure 12b). Moreover, after UV-B light treatment, the score for sweetness decreased by 10%, while the bitterness score increased by 11%. However, the overall assessments of UV-B-treated lettuce cultivars were slightly increased (3.9 to 4.1 for both), presumably due to improved appearance and crispness properties. On the contrary, UV-C exposition significantly decreased the overall assessment by 31 and 39% for LB and LR, respectively. The lowest scores were noted, especially in the appearance category (Figure 12a,b).

## 3. Discussion

### 3.1. Efficiency of RGB Spectrum Supplementation With UV-B or UV-C Light on Antioxidant Capacity, Morphology and Sensory Properties of Lettuce Cultivars

The consumption of produced fruits and vegetables is strictly associated with the prevention of many diseases due to the antioxidant activity of plant secondary metabolites [18]. However, the anticipated expansion of indoor farming, which employs strictly controlled and stable growing conditions, may restrict the levels of health-promoting compounds, as these compounds typically accumulate in response to abiotic stresses. In this area of study, UV radiation is an especially underrated factor currently missing in most horticultural lighting systems of plants applied to protected cultivation, such as consistent (indoor farming) and inconsistent (greenhouse) systems [21]. Therefore, this study aims to identify an efficient, easy-to-operate, and non-invasive method to biofortify plant tissue with secondary metabolites for use in indoor farming. The proposed method involves the short-term exposure of plants to low doses of UV radiation directly before harvest. In our study, a short-term assay instead of a long-term treatment was applied, as the results of previous research [10] clearly indicated that prolonged UV-light exposition, despite enhancing the bioactive compound level, also exerted a negative impact on biomass yield. Moreover, the lettuce plants, that are biofortified with UV light are intended to be harvested after a short-grown cycle in the form of *baby leaf*, as previous research [9] documented that the amount of antioxidants declines with plant age. A short production cycle might also be a beneficial strategy for minimizing operational costs and increasing the durability of artificial lighting. To this end, we employed LED-based RGB lighting, but the real breakthrough for UV supplementation will also be LED-based UV-B systems in place of the mercury lamps used in the present study. 

However, the presented assay has already been successfully employed in our previous study [22] for the pre-harvest UV-A light (365 nm) supplementation to the RGB spectrum to enhance antioxidant accumulation in lettuce and basil plants. Thus, in this paper, we further analyzed the influence of UV light from the UV-B and UV-C regions on the antioxidant properties of a popular leafy vegetable, lettuce, grown in both green and reddish cultivars. Moreover, as the employed UV wavelengths are characterized by high-energy photons, the impact of UV light treatment has also been studied, and it is discussed in the next chapter. 

A similar approach has been previously analyzed in research [23], which evaluated the effect of various UV wavelengths of exposition on phenolic compound accumulation, growth, and photosynthetic activity of red leaf lettuce cv. Hongyeom. The mentioned authors applied UV light in time-constant, repeated doses (6 days, 4 h per day for UV-B, 3 days, 2 h per day for UV-C) or with daily increasing doses (from 1 to 7 h per day, 6 days, UV-B) and documented that UV-B (306 nm) or UV-C (253.7 nm) treatment increased the total phenolic concentration by 3.6 and 3.2 times, respectively, compared with control. There was, however, no significant difference between the accumulation of phenolics with constant and gradually increasing doses of UV-B light. The authors also noted enhanced antioxidant capacity of lettuce measured with the ABTS assay. 

In the case of our study, we applied a narrowband UV-B lamp, typically used for phototherapy [24], with a dominant peak around 311 nm and a minor one at 364 nm. Analyses proved that 4-day-long progressive supplementation of UV-B light to the RGB background applied prior to harvest is a sufficient and effective method for increasing the total phenolic (TPC) (Figure 1), flavonoid (TFC) (Figure 2), and anthocyanins (ANT) (Figure 3) content, both in green and red leaf lettuce cultivars. UV-B light exposure has been documented to be the most efficient way to increase TFC and ANT. Moreover, in the case of TPC and TFC green leaf cultivars, Lollo Bionda has been proven to be significantly more susceptible to enhancing their accumulation in response to UV-B than cv. Lollo Rossa. An explanation for such results is the fact that the reddish cultivar is already characterized by 3, 2.7, and 5.5 times higher levels of TPC, TFC, and ANT under the control spectrum, respectively, compared to cv. Lollo Bionda. Yet, UV-B exposure significantly enhanced the content of ANT levels in LR plants (Figure 3b). Also, the case of the ascorbic acid (AsA) study proved that UV-B exposure was more efficient in inducing its accumulation in green leaf lettuce, while this effect has been mostly attributed to the accumulation of the oxidized form of AsA (Figure 4b). Consequently, UV-B exposure enhanced the overall antioxidant capacity (TAC) of green leaf cultivar extracts (Figure 5) by two-thirds and presented 10% higher radical scavenging activity compared to plants grown solely under the RGB spectrum. In the case of the LR cultivar, no further increase in TAC has been noticed, despite UV-B exposure, due to the initial high level of antioxidant compounds. Also, in our previous study [22], we documented that LB lettuce presents significantly higher responsiveness to UV-A-dependent phenolic compound synthesis than reddish LR plants. In leaves, phenolic accumulation protects the photosynthetic apparatus against UV damage; thus, the green cultivars presented significantly lower TPC when grown without stressors such as UV, while making them more vulnerable to UV light exposition that activates secondary metabolite synthesis and deposition. The mechanism underlaying TPC, TFC, and ANT synthesis in response to UV light may be attributed to its ability to induce the gene expression of phenylalanine ammonia lyase (*PAL*), a key enzyme involved in the first step of the phenylpropanoid pathway [25]. 

Still, however, UV-B exposure also induced carotenoid accumulation in LR plants (Table 2), indicating its application might also be considered as a tool for the biofortification of plant products with other phytochemicals. Carotenoids, like phenolics and AsA, present antioxidant activity and protect cells and tissues from damage by free radicals and singlet oxygen, providing enhancement of immune function, protection from sunburn reactions, and delaying the onset of certain types of cancer [26]. Although no induction of carotenoids level has been observed for the LB cultivar, previous research [27], which tested other green and red leaf lettuce cultivars, noted that UV-B supplementation increased the carotenoids pool of green leaf lettuce while reducing the levels of these compounds in the red leaf plants. Thus, the reaction might be cultivar- or dose-dependent, as has been shown in a previous paper [28]. Moreover, as previous research [29] documented that changes in bioactive phytochemicals influence the sensory properties (i.e., bitterness, aftertaste) of lettuce, we also conducted consumers’ tests and showed that UV-B exposure in fact decreased sweetness and aftertaste scores while increasing the bitterness rate for both cultivars. At the same time, however, UV-B treatment improved the appearance and crispness; thus, the overall assessment for UV-B-treated lettuce was a bit higher than for RGB (Figure 12). 

In the study, the influence of UV-C short-term exposition has also been analyzed. The dominant peak in the spectrum of the analyzed UV-C lamp is at 253/254 nm, with a minor peak at 312/313 nm. However, as the UV-C light is high-energy radiation, we have chosen a 254 nm UV-C lamp, which has been shown to exert a less negative impact on plant tissue than 222 nm lamps [30] and not produce ozone. The employed UV-C low-pressure mercury lamp has been equipped with a doped quartz envelope to block 185 nm radiation emission, minimizing O_3_ generation [31] and the risk of non-specific harmful effects of ozone stress [32]. Moreover, we also adjusted the times of individual exposition according to the previous study [23], reducing the cumulative dose of UV-C to 6.008 kJ m^−2^, compared to 15.622 kJ m^−2^ of UV-B. Results showed that in the LB cultivar, the UV-C light exposure exerted no effect on TPC (Figure 1a) or AsA (Figure 4a), while increasing TFC (Figure 2a) and ANT (Figure 3a) by nearly 2 and 1.5 times, respectively. Consequently, in the case of Lollo Bionda lettuce, the TAC parameter increased by 53% (Figure 5a), indicating that UV-C is a potent factor in increasing the antioxidant capacity of green lettuce. However, following the UV-C exposure, the LB presented reduced leaf areas and severe morphological trait changes that included leaf glazing, bronzing, and curling (Figure 10c), as has also been noted in previous research [14]. In the case of red leaf lettuce, we showed the UV-C light exerted a negative impact on the analyzed phenolic compounds and carotenoid content while increasing only slightly the total AsA level. Thus, as a result of UV-C exposure, LR plants presented significantly reduced TAC as well as radical scavenging activity (Figure 5b,d). In addition, UV-C treatment of LR, similar to effects noted for LB, induced the occurrence of negative morphological traits (Figure 11c) and reduced the scores of consumers’ sensory tests within the appearance, crispness, and overall assessment categories (Figure 12a,b). On the contrary, researchers documented that UV-C in low doses is an elicitor of the biosynthesis of carotenoids and flavonoids in red bell peppers [33] and in tatsoi baby leaves combined with hyperoxia conditions [34].

The discrepancies in UV-B and UV-C efficiency to induce the synthesis of antioxidants and their impact on plants’ conditions are related to their modes of perception and action. Upon UV-B exposure, inactive UVR8 dimers monomerize in the cytoplasm and accumulate in the nucleus. UVR8 monomers are crucial to avert the breakdown of the HY5 transcription factor. HY5 accumulation induces flavonoid biosynthesis by upregulating genes such as chalcone synthase (*CHS*), flavonol synthase (*FLS1*), and chalcone flavanone isomerase (*CHI*) [14]. UV-C, however, is considered to be perceived by plants through redox state change, as its exposition induces ROS production through the mitochondrial electron transport chain and NADPH oxidase [35]. Moreover, analysis [35] suggests that UV-C helps in the retention of AsA and phenolic content in acerola by altering ascorbic acid and phenolic metabolism. The authors noted that UV-C activated L-galactono-1,4-lactone dehydrogenase (GalDH), a key enzyme for vitamin C biosynthesis, and altered the composition of phenolic compounds through phenolic biosynthesis. Such conclusions are consistent with our results obtained for green leaf cultivars with lower initial levels of SM, but we do not agree when analyzing the reddish lettuce. Thus, it might be that such an effect is a consequence of the continuous quenching of ROS with antioxidant phytochemicals. An increased accumulation of highly absorbing UV compounds such as phenolics and flavonoids causes more UV radiation to be absorbed and generates ROS that exceeds the ability of plants to scavenge, followed by ROS-related damages. Alternatively, observed UV-C induction of phytochemical synthesis might be related to UVR8 activation, as it is postulated that UV radiation that exceeds 250 nm might be absorbed within UVR8, thus sharing the same signaling pathway as UV-B light [13].

### 3.2. Condition of Photosynthetic Apparatus in Response to UV-B or UV-C Supplementation to the RGB Spectrum

The adverse effects of UV radiation on the structure and function of the photosynthetic apparatus (PA) are well known and documented [36]. While, however, the effects of UV-A radiation may be damaging or non-damaging, even mitigating the deleterious effects of other UV wavelength regions [37], UV-B and especially UV-C light action are mostly considered to have adverse effects on photosynthesis efficiency. UV-B irradiation has been shown to decrease the amount of plastoquinones (PQs) and impair their function in the PSII complex. Moreover, the prime action sites of UV-B that contribute to the decrease in photosynthetic activity are CO_2_ fixation and oxygen evolution with Mn cluster, impairment of PSII by damaging of D1/D2 reaction center proteins, and to a lesser extent, of PSI proteins, reduction of total chlorophyll, Rubisco content and activity, and inactivation of ATP synthase [36,38]. However, when applied at low doses, UV-B radiation has been documented to not necessarily have a damaging effect on photosynthesis or pigment level, and the treatment is mostly not lethal as PA readily recovers. Moreover, analyses of different corn hybrids clearly documented that there is a significant variation in resistance to the adverse effect of UV-B radiation between plants, and the relative change in photosynthesis can be used as a measure of their resistance to the harmful effect of UV-B [39]. Moreover, as consistent with our results, the UV-B light induced a shift in antioxidant balance towards the synthesis of UV-absorbing pigments such as flavonoids and carotenoids and increased the stress resistance of the PA [40]. 

However, in the case of UV-C, it possesses an indisputable adverse effect on photosynthesis, which is related mostly to higher-energy photons of the UV-C wavelength that induce rather destruction than impairment of PA’s structures, as reported for PQs [36]. However, little is known about the exact mechanism underlying the UV-C-related photosynthesis impairment. In a review [41], the negative effect of UV-C treatment on the PA of lettuce is linked to its damaging effect on PQs, which has been recorded with a fluorescence induction curve. Similarly, we also observed an almost immediate (day 1) drop in photosynthetic activity in both lettuce cultivars after UV-C treatment (Figure 7a and Figure 8a). Moreover, UV-C exposition is also proposed to exert a damaging effect on the integrity of thylakoids as they undergo fusion and the accumulation of starch. Also, the part of the UV-C between 254 and 262 nm spectrum is the most effective for DNA and protein molecules damage and inhibits mitochondria and chloroplast activity due to the production of ROS [41]. Analyses of chlorophyll *a* fluorescence parameters in previous research [41] showed that *Arabidopsis thaliana* plants exposed to low-dose UV-C light presented reductions in Fv/Fm, ΦPSII, and NPQ in a similar manner to LB and LR lettuce tested in this paper. Interestingly, a study [42] stated that phot1 and phot2 (phototropin) receptors contribute to the inhibition of UV-C-induced foliar cell death. In such an explanation, UV-C treatment decreases the expression of *PHOT1* and *PHOT2* genes, while genes of light-harvesting complexes (*LHCB1.1*, *LHCB2.1*, *LHCB2.2,* and *LHCB2.4*) were significantly upregulated after treatment, presumably as a result of blocked phototropin-dependent signaling. Thus, it might be an explanation for the observed increased SLP and LSU/SSU in LB cultivars in RGB + UV-C plants. Also, another study [43] showed that both UV-B and UV-C light exposition increased the soluble protein content in the leaves of *Tetrastigma hemsleyanum*. 

In such a scenario, LR plants that presented a higher initial content of UV-C absorbing phytochemicals [44] are less vulnerable to the UV-C effect, both positive, related to SM accumulation, and adverse, related to PA disruption, as proved with analyses of TBARS. Additionally, the lower negative impact of UV-C on LR, compared to LB, might have contributed to a higher accumulation of AsA. Previous research [45] documented a UV-protective effect on PSII centers in grana and stroma lamellae after exogenous application of AsA in pea seedlings (*Pisum sativum* L. cv. Borec), related to Mn cluster protection and stabilization of PSII rather than to the direct increase in ROS scavenging activity. Such a conclusion seems to also be genuine for our analysis, as in the LR cultivar, increased total AsA levels showed no correlation to the DPPH scavenging rate. 

Taken together, our results are in accordance with previous studies [23,46] and showed that UV-C light induces the accumulation of phytochemicals, especially in green leaf cultivars, although it provokes negative morphological changes and inhibits plant growth. Consequently, pre-harvest UV-C application to improve nutraceutical properties in ready-to-eat leafy vegetables is limited. Previous research documented, however, that UV-C could be considered an effective tool for postharvest application, especially in less fragile plant foods, such as grains, as analyses [15] demonstrated that UV-C irradiation is an effective method for fungal control and reducing mycotoxins in stored rice.

## 4. Materials and Methods

### 4.1. Plant Material, Growth Conditions and Light Treatment

*Baby leaf* lettuce (*Lactuca sativa* var. *crispa* L.) cultivars with green (cv. Lollo Bionda, LB) and reddish leaves (cv. Lollo Rossa, LR) were sown in P9 containers (9 × 9 × 10 cm) filled with a substrate composed of peat, perlite, and an N–P–K ratio of 9:5:10 (pH = 6.0–6.5). The containers were divided into groups and transferred to environmentally controlled growth chambers. The plants were cultivated for 20 consecutive days (20 DAS, days after sowing) under LED RhenacM12 lamps (PXM, Podleze, Poland) providing 200 µmol m^−2^ s^−1^ of the RGB spectrum (R–G–B; 661:633:520:434 nm) applied solely (control, Figure 13a) or under the RGB spectrum supplemented 4 days prior to harvest with increasing doses of UV-B (311 nm, PL-S 9W/01/2P 1CT/6X10BOX, Philips Lighting, Eindhoven, The Netherlands, Figure 13b) or UV-C (254 nm, TUV PL-S 9W/2P 1CT/6X10BOX, Philips Lighting, Figure 13c), aligned with the schedule presented in Table 3.

The RGB treatment served as the control group. Light composition and photosynthetic photon flux density (PPFD) were monitored daily using a calibrated spectroradiometer, GL SPECTIS 5.0 Touch (GL Optic Lichtmesstechnik GmbH, Weilheim/Teck, Germany). The containers with plants were turned in twice a day. The photoperiod was 16/8 h (day/night; day 6.00 am–10.00 pm), the average air temperature was maintained at 23/20 °C (day/night), the relative air humidity was kept at 50–55%, and CO_2_ was maintained at 420 ± 10 µmol mol^−1^. The plants were watered with tap water when necessary and fertilized once a week with 1% (*w*/*v*) fertilizer (N–P–K = 9:9:27; Substral Scotts, Warszawa, Poland). Ten plants (two repetitions with five plants per light condition) were grown with each kind of light treatment.

### 4.2. Estimation of Total Phenolic Content with Folin–Ciocalteu Assay

Estimation of total phenolic content (TPC) was conducted, as described earlier [47]. In brief, 100 mg of fresh weight (FW) leaf tissue was placed in tubes with 1.0 mL of methanol. Samples, kept in dim light, were vortexed for 20 s and incubated for 30 min at 60 °C with inversion every 10 min to improve extraction. Then, the sample mixture was centrifuged at 10,000× *g* for 2 min, and then the supernatant was carefully collected without disturbing the plant tissue, transferred to a new tube, and mixed once again for 15 s. Then 100 µL of each extract, cooled down to room temperature (RT), was mixed with 200 µL of 10% (*v*/*v*) Folin–Ciocalteu reagent (F-C) and vortexed twice for 10 s. Then, 800 µL of 700 mM Na_2_CO_3_ was added, vortexed twice for 10 s, and incubated for 30 min at 40 °C, protected from light. After incubation, the mixture was centrifuged at 10,000× *g* for 1 min and transferred to a 96-well microplate with 200 µL per well. For TPC determination, the absorbance (Abs) at 765 nm was estimated with a microplate spectrophotometer (Mobi, MicroDigital Co., Ltd., Seongnam, Republic of Korea) with six replicates. The standard curve with gallic acid (0–200 nmol) was used to estimate the phenolic compound amount (gallic acid equivalents) in a sample, and expressed as µg per mg of FW.

### 4.3. Estimation of Total Flavonoid Content

For the measurement, the total flavonoid (TFC) assay [48] with modification was applied. The 60 µL of methanol extract obtained previously for the TPC assay was mixed with 680 µL of 30% (*v*/*v*) methanol–water and 30 µL of 0.5 M NaNO_2_, vortexed for 20 s, and incubated at RT for 3 min without light. Then 30 µL of 0.3 M AlCl_3_ × 6H_2_O was added to each sample, vortexed for 20 s, incubated at RT for 3 min, and then mixed with 200 µL of 1 M NaOH, vortexed, and left for the next 40 min at RT without light. After incubation, samples were mixed, shortly centrifuged (5000× *g* for 1 min), and 200 µL aliquots of each sample were transferred to a 96-well microplate. For TFC determination, the Abs at 506 nm was estimated with a microplate spectrophotometer with six replicates. The flavonoid content in the sample extracts was quantified using calibration curves of flavonoid standards for rutin.

### 4.4. The Ascorbate/Dehydroascorbate (AsA/DAsA) Ratio Estimation

Ascorbic acid (AsA) was determined by the bipyridyl method [49]. The ascorbate/dehydroascorbate (AsA/DAsA) ratio is an indicator of the stress level in plants. The method involves the extraction and determination of AsA and DAsA. The assay is based on the reduction of Fe^3+^ to Fe^2+^ by AsA and the spectrometric determination of Fe^2+^ in complex with 2,2′-bipyridyl. DAsA is reduced to AsA by pre-incubation of the sample with dithiothreitol (DTT) dissolved in 0.2 M phosphate buffer (Na_2_HPO_4_/NaH_2_PO_4_) at pH = 7.4. The excess DTT has been removed using N-ethylmaleimide (NEM, Sigma), and the total AsA concentration is determined using the 2,2′-bipyridyl method. The DAsA concentration is assessed by the difference between the total and initial AsA concentrations. 

In brief, 500 mg of plant samples was homogenized into a fine powder in a mortar placed on ice with the addition of 1.5 mL of 6% TCA. The homogenate was transferred to a 2 mL tube and centrifuged for 5 min at 15,000× *g* (4 °C). The supernatant was transferred to a tube and immediately analyzed for AsA and DAsA presence. Absorbance was read at 525 nm. L-ascorbic acid solutions in concentrations of 0, 0.06, 0.125, 0.25, 0.5, and 1.0 µM dissolved in 6% (*w*/*v*) TCA were used to determine the calibration curve for AsA. The analysis was performed in six replicates for each treatment.

### 4.5. Anthocyanins Assay

The levels of anthocyanins (ANT) were measured, as described earlier [50]. Plant tissue (200 mg) was extracted with 1 mL methanol–HCl (99:1, *v*/*v*) at 4 °C. The sample’s Abs was spectrophotometrically measured at 530 and 657 nm with six replicates, and the relative anthocyanin levels [AU g^−1^ FW] were determined using Equation (1):(1)$$\frac{{Abs}_{530}-\left(0.25\times {Abs}_{657}\right)\times extraction\;volume\;\left(mL\right)\times 1}{Mass\;of\;tissue\;sample\;\left(g\right)}=\frac{Relative\;units\;of\;anthocyanins\;}{g\;Fresh\;weight\;of\;plant\;tissue}$$

### 4.6. Antioxidant Activity by DPPH Assay

The antioxidant activity of each plant extract was measured by the 1,1-diphenyl-2-picrylhydrazil (DPPH) scavenging assay, according to the previous study [51]. For the DPPH assay, the 60 µL of plant methanol extract obtained previously for the TPC assay was mixed with 904 µL of methanol and 576 µL of 0.125 mM DPPH in methanol, vortexed for 20 s, and incubated for 30 min at 37 °C. Using a microplate spectrophotometer, the Abs of each sample was measured at 517 nm with six replicates. To determine sample radical scavenging activity, the calibration curve with a synthetic antioxidant—butylated hydroxytoluene (BHT) (0–400 µg per mL) and 0.125 mM DPPH—was plotted. The following formula was used to calculate the percentage of DPPH scavenging activity (2):(2)$$DPPHinhibition\left(\%\right)=\frac{Absorbance\;of\;control*-Absrobance\;of\;sample}{Absorbance\;of\;control}\times 100$$

* control states for DPPH mixture incubate with 0 µg BHT solution.

### 4.7. Photosynthetic Pigments Determination

The concentrations of chlorophyll *a* and *b* and total carotenoids were measured spectrophotometrically after being dissolved in dimethyl sulfoxide (DMSO). Pigments were extracted from approximately 20 mg of leaf tissue in 1.0 mL of DMSO. Samples, kept in dim light, were vortexed for 1 min, then capped and incubated for 3 h at 65 °C with inversion every 10 min to improve extraction. Then the sample mixture was centrifuged at 10,000× *g* for 5 min, and the supernatant was carefully collected and transferred to a new tube. Pigment determination was carried out according to previous assays [52] at 480, 649, and 665 nm, with formulas suitable for 1 nm resolution. The assay was performed in six replicates for each treatment.

### 4.8. Leaf Soluble Protein Level and Densitometric Analysis of RuBisCO Subunits

Soluble leaf proteins (SLPs) were extracted with an alkaline lysis method according to the previous procedure [53]. Plant material was incubated for 10 min at 90 °C in 500 µL of alkaline lysis buffer (0.1 M NaOH, 0.05 M EDTA, 2% SDS, and 2% β-mercaptoethanol). After cooling to RT, 5 µL of 4 M acetic acid was added. The tubes were then vortexed and incubated again for a maximum of 10 min at 90 °C. The obtained supernatant was used to assess protein content using a NanoDrop 2000 spectrophotometer (Thermo Fisher Scientific, Waltham, MA, USA) at a wavelength of 280 nm.

Then, the calculated amount of each extract mixed with Laemmli Sample Buffer (Bio-Rad, Hercules, CA, USA) was loaded onto precast 4–20% gradient TGX polyacrylamide gels (Bio-Rad) and run with a constant voltage of 200 V for 20 min. Three replicates of each treatment were analyzed. Gels were stained with Bio-Safe™ Coomassie Stain (CBB, Bio-Rad). The quantification of the protein bands of the CBB-stained gels was made using densitometric analysis (ImageJ v. 1.52, National Institutes of Health, Bethesda, MD, USA). The relative amount of RuBisCO subunits was calculated using, as a maximum, the value measured in RGB control plants [54].

### 4.9. Measurement of Chlorophyll Fluorescence (ChF) Induction Kinetics

ChF induction kinetics of control and UV-treated lettuce leaves were performed using a pulse amplitude modulated (PAM) fluorometer (Maxi IMAGING-PAM M-Series, Walz, Effeltrich, Germany). The minimal fluorescence level (Fo) with all PSII reaction centers open was measured by measuring modulated blue light (λ = 450 nm, 0.01 μmol m^−2^ s^−1^). The maximal fluorescence level (Fm) with all PSII reaction centers closed was determined by a 0.8 s saturating pulse at 2700 μmol m^−2^ s^−1^ in 30 min dark-adapted leaves. Then, the leaf was continuously illuminated with blue actinic light (186 μmol m^−2^ s^−1^). The maximum quantum yield of PSII (Fv/Fm), actual photochemical efficiency of PSII (ΦPSII), quantum yield of regulated energy dissipation in PSII (ΦNPQ) and non-regulated energy dissipation in PSII (ΦNO), non-photochemical energy quenching (NPQ), and electron transport rate in PSII (ETR) were measured every other day after UV light exposition. All analyses were conducted between 8.00 am and 10.00 pm.

### 4.10. Measurement of Lipid Peroxidation Rate

The level of oxidative damage to membranes in response to UV treatment was estimated indirectly with an assessment of byproducts of lipid peroxidation reacted with thiobarbituric acid (TBA), among them malondialdehyde (MDA) content. The assay was in accordance with the previous procedure [55]. In brief, 200 mg of leaf tissue was homogenized in 1 mL of methanol and incubated at 60 °C for 30 min. After centrifugation at 10,000× *g* for 5 min, 300 µL of each extract was mixed with 600 µL of the TCA-BHT-TBA mixture with 0,18 M, 65,5 µM, and 45 mM, respectively. Mixed tubes were incubated for 5 min at 95 °C. After centrifugation at 10,000× *g* for 1 min, the Abs of supernatants was measured at 532 nm, with values corresponding to non-specific absorption at 450 nm and a correction factor for non-specific turbidity at 600 nm. The MDA concentration [μmol g^−1^ FW] determined on a fresh weight basis was calculated according to previous research [54] with the following Formula (3):(3)$$\;MDA=6.45\times \left({Abs}_{532}-{Abs}_{600}\right)-0.56\times {Abs}_{450}$$

### 4.11. Sensory Analysis

The sensory parameters of the fresh (raw) leaf samples of lettuce: appearance, sweetness, bitterness, crispness, aftertaste, and overall assessment were scored on a scale of 0–5 (where 5 is maximal, 3—neutral, and 0—the least scored) by 10 untrained consumers, aged 25–65 years (equal gender ratio) in accordance with the previous procedure [56]. The leaf samples were randomly selected, and the principle of “sample blinding” was applied to minimize bias among consumers.

### 4.12. Models for Data Fitting and Statistical Analysis

The fitting of experimental data on DPPH inhibition by BHT used for DPPH radical scavenging activity rate was performed using OriginPro version 2024 (OriginLab Corporation, Northampton, MA, USA).

Statistical analyses were performed using Statistica 13.3 software (StatSoft Inc., Oklahoma, OK, USA). The normal distribution of variables was verified using the Shapiro–Wilk test, and the equality of variances was evaluated using Levene’s test. One-way ANOVA and post hoc Tukey’s HSD tests were employed to analyze the differences between the investigated groups. The data are presented as the mean with standard deviation (±SD). Statistical significance was determined at the 0.05 level (*p* = 0.05).

## 5. Conclusions

The results demonstrate that low-dose short-term UV-B supplementation (4 days, total 3.75 h) in a red–green–blue light spectrum allows for the induction of the accumulation of health-promoting phytochemicals such as phenolics, flavonoids, anthocyanins, carotenoids, and ascorbic acid and enhances the overall antioxidant capacity, especially in green leaf cultivars of lettuce (Lollo Bionda type), while in reddish leaf cultivars (Lollo Rossa type), the positive effect of UV-B supplementation on antioxidant accumulation is more limited, presumably due to the initial higher concentration of protective compounds. Analyses also showed that UV-B application at a cumulative dose of 15.622 kJ m^−2^ showed no adverse effect either on the photosynthetic activity or morphology of both lettuce cultivars. On the contrary, UV-C supplementation, despite lower doses applied (CD = 6.008 kJ m^−2^), is effective only in green leaf cultivars, where it successfully induced antioxidant accumulation. However, even despite that, its application for lettuce biofortification is not recommended as it exerts an adverse effect on both cultivars’ morphology, with leaf glazing, bronzing, and curling, thus significantly decreasing the quality of the leafy products obtained, as confirmed by consumers’ sensory tests. In the case of reddish leaf Lollo Rossa, we noted that plants were less vulnerable to UV-C treatment, but it had both positive and negative outcomes, as better photosynthetic apparatus protection was accompanied by minor activation of phytochemical synthesis. Taken together, our study provides important solutions for indoor lighting improvement, mostly devoid of lamps emitting UV radiation, especially in the UV-B region. However, as the effectiveness of applied short-term low-dose UV-B light for green and reddish lettuce cultivars was different, further analyses are needed to clarify whether such a response is also reliable for other lettuce cultivars as well as other leafy vegetables such as basil or kale. Moreover, to address the limitation of UV-B light inducing phytochemical synthesis within the more pigmented cultivars, the extension of time/dose of applied UV light should also be considered in future research.

## Figures and Tables

**Figure 1 ijms-25-09298-f001:**
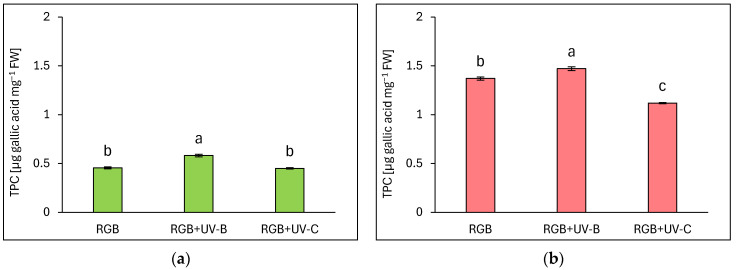
Total phenolic content (TPC) of control (RGB), UV-B treated (RGB + UV-B), or UV-C treated (RGB + UV-C) plants of *baby leaf* lettuce (*Lactuca sativa* var. *crispa* L.) cultivar with (**a**) green (cv. Lollo Bionda) and (**b**) reddish leaf (cv. Lollo Rossa) at 20 DAS (days after sowing), estimated as µg gallic acid equivalents per mg of fresh weight (FW). Each bar represents the average ± SD of six independent measurements (*n* = 6). Different letters (a–c) indicate significant differences between treatments at *p* = 0.05 with a Tukey’s HSD test.

**Figure 2 ijms-25-09298-f002:**
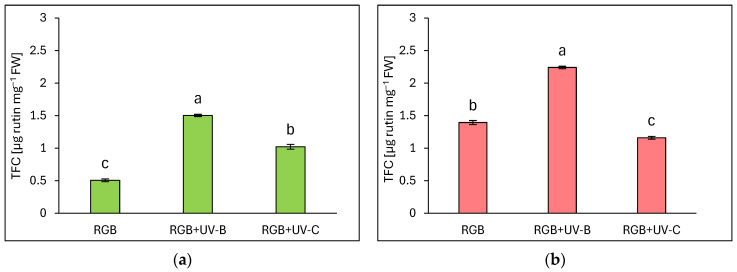
Total flavonoid content (TFC) of control (RGB), UV-B treated (RGB + UV-B), or UV-C treated (RGB + UV-C) plants of *baby leaf* lettuce (*Lactuca sativa* var. *crispa* L.) cultivar with (**a**) green (cv. Lollo Bionda) and (**b**) reddish leaf (cv. Lollo Rossa) at 20 DAS (days after sowing), estimated as µg rutin equivalents per mg of fresh weight (FW). Each bar represents the average ± SD of six independent measurements (*n* = 6). Different letters (a–c) indicate significant differences between treatments at *p* = 0.05 with a Tukey’s HSD test.

**Figure 3 ijms-25-09298-f003:**
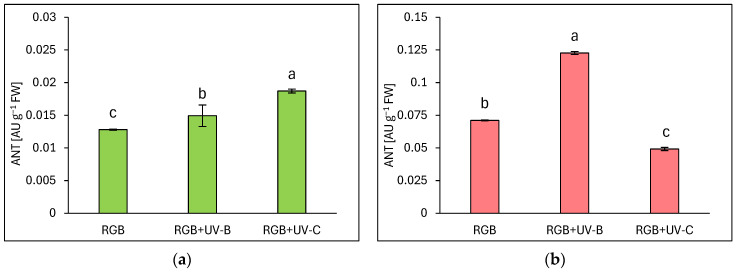
Anthocyanins (ANT) concentration of control (RGB), UV-B treated (RGB + UV-B), or UV-C treated (RGB + UV-C) plants of *baby leaf* lettuce (*Lactuca sativa* var. *crispa* L.) cultivar with (**a**) green (cv. Lollo Bionda) and (**b**) reddish leaf (cv. Lollo Rossa) at 20 DAS (days after sowing), estimated as arbitrary unit (AU) per g of fresh weight (FW). Each bar represents the average ± SD of six independent measurements (*n* = 6). Different letters (a–c) indicate significant differences between treatments at *p* = 0.05 with a Tukey’s HSD test.

**Figure 4 ijms-25-09298-f004:**
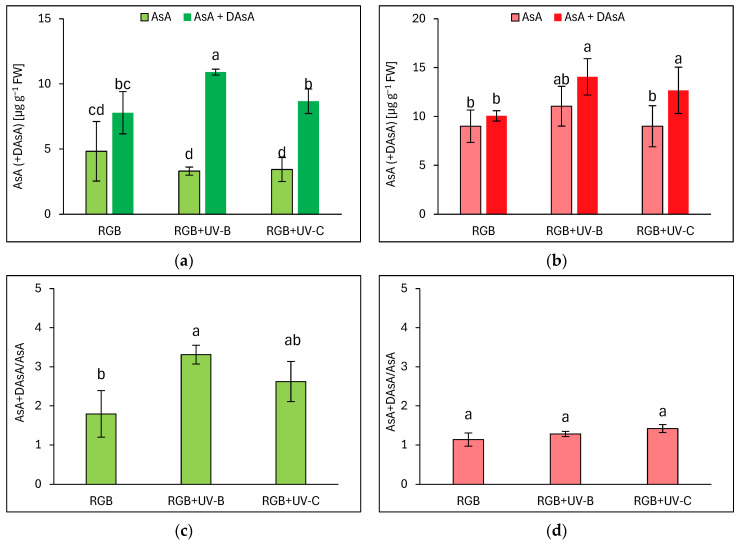
Initial ascorbic acid (AsA) level, total AsA pool (AsA + DAsA), and total AsA to initial AsA ratio (AsA + DAsA/AsA) of control (RGB), UV-B treated (RGB + UV-B), or UV-C treated (RGB + UV-C) plants of *baby leaf* lettuce (*Lactuca sativa* var. *crispa* L.) cultivar with (**a**,**c**) green (cv. Lollo Bionda) and (**b**,**d**) reddish leaf (cv. Lollo Rossa) at 20 DAS (days after sowing). Initial AsA was estimated directly in a sample by bipyridyl method, while the total AsA pool was assessed after additional reduction of dehydroascorbic acid (DAsA) with dithiothreitol (DTT). Each bar represents the average ± SD of six independent measurements (*n* = 6). Different letters (a–c) indicate significant differences between treatments at *p* = 0.05 with a Tukey’s HSD test.

**Figure 5 ijms-25-09298-f005:**
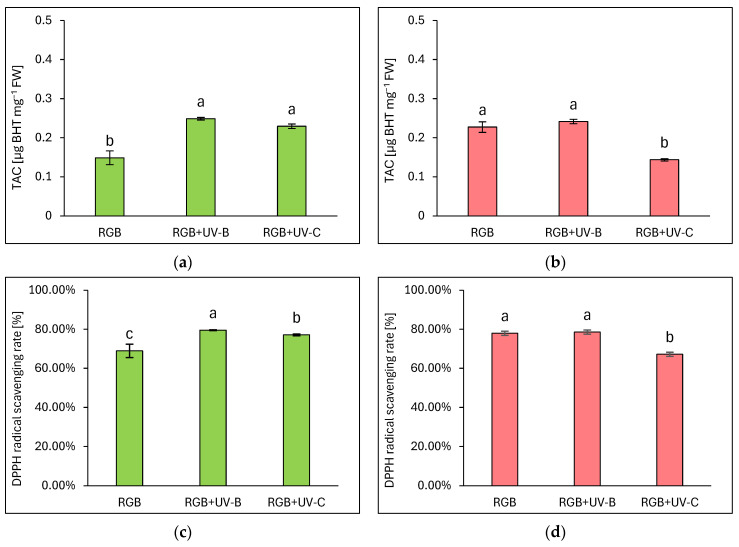
The total antioxidant capacity (**a**,**b**) and DPPH radical scavenging activity rate (**c**,**d**) of control (RGB), UV-B-treated (RGB + UV-B), or UV-C-treated (RGB + UV-C) plants of *baby leaf* lettuce (*Lactuca sativa* var. *crispa* L.) cultivar with (**a**,**c**) green (cv. Lollo Bionda) and (**b**,**d**) reddish leaf (cv. Lollo Rossa) at 20 DAS (days after sowing), estimated as µg BHT equivalents per mg of fresh weight (FW). Each bar represents the average ± SD of six independent measurements (*n* = 6). Different letters (a–c) indicate significant differences between treatments at *p* = 0.05 with a Tukey’s HSD test.

**Figure 6 ijms-25-09298-f006:**
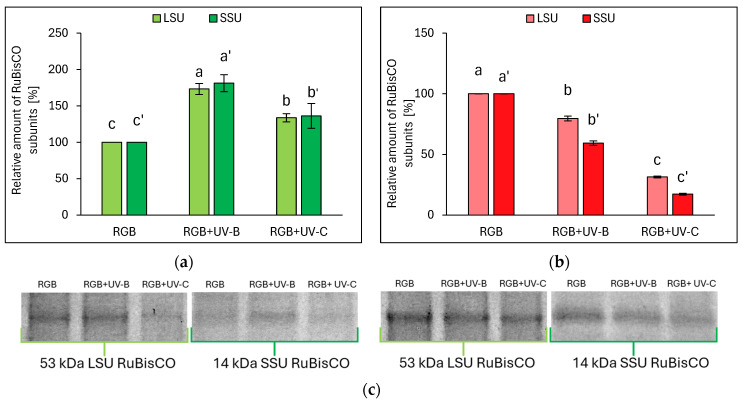
Densitometric analysis of RuBisCO large (LSU) and small (SSU) subunit of control (RGB), UV-B-treated (RGB + UV-B), or UV-C-treated (RGB + UV-C) plants of *baby leaf* lettuce (*Lactuca sativa* var. *crispa* L.) cultivar with (**a**) green leaf (cv. Lollo Bionda) or (**b**) reddish leaf (cv. Lollo Rossa) after short-term (1–4 day) progressive exposition to UV light at 20 DAS (days after sowing). Beneath (**c**) the LSU (53 kDa) or SSU (14 kDa) protein bands of leaf proteins resolved in a 4–20% TGX polyacrylamide gel and visualized with Coomassie Stain. The relative amounts (%) of RuBisCO subunits were normalized to RGB control. Bars represent the average ± SD of three independent measurements (*n* = 3). Different letters (a–c for LSU or a’–c’ for SSU) indicate significant differences between treatments at *p* = 0.05 with a Tukey’s HSD test.

**Figure 7 ijms-25-09298-f007:**
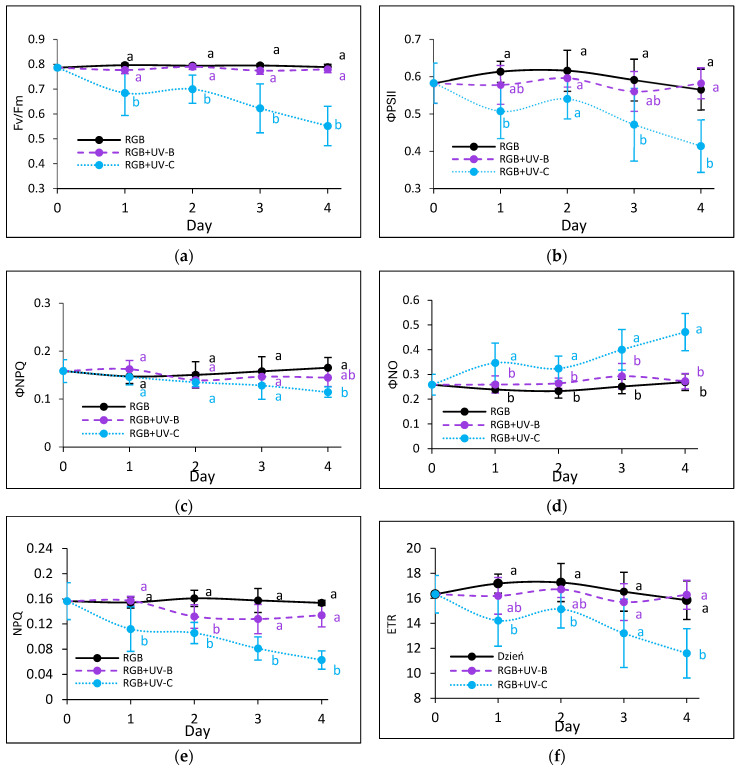
Chlorophyll *a* fluorescence analysis of control (RGB), UV-B-treated (RGB + UV-B) or UV-C-treated (RGB + UV-C) plants of *baby leaf* lettuce (*Lactuca sativa* var. *crispa* L.) cultivar with green leaf (cv. Lollo Bionda) after short-term (1–4 day) progressive exposition to UV light. (**a**) The maximum quantum yield of PSII photochemistry (Fv/Fm), (**b**) effective quantum yield of PSII photochemistry (ΦPSII), (**c**) quantum yield of regulated (ΦNPQ), (**d**) non-regulated energy dissipation (ΦNO), (**e**) non-photochemical quenching (NPQ), and (**f**) electron transport rate (ETR). The analyses were carried out with 55 μmol m^−2^ s^−1^ of blue (450 nm) actinic light. Each data point represents the average ± SD of six independent measurements (*n* = 6). Different letters (a, b) indicate significant differences between treatments at *p* = 0.05 with a Tukey’s HSD test.

**Figure 8 ijms-25-09298-f008:**
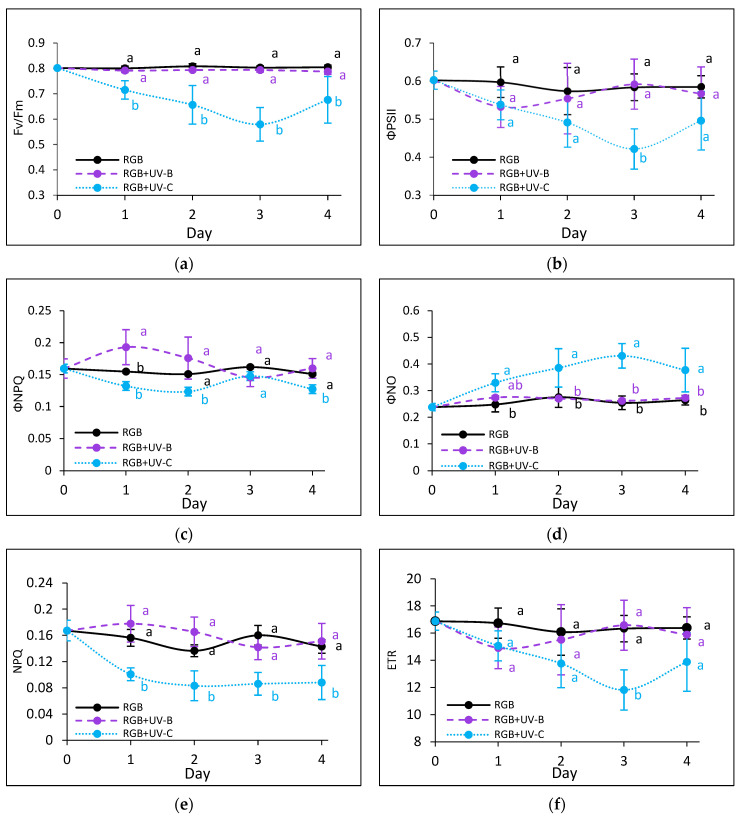
Chlorophyll *a* fluorescence analysis of control (RGB), UV-B-treated (RGB + UV-B), or UV-C-treated (RGB + UV-C) plants of *baby leaf* lettuce (*Lactuca sativa* var. *crispa* L.) cultivar with reddish leaf (cv. Lollo Rossa) after short-term (1–4 day) progressive exposition to UV light. (**a**) The maximum quantum yield of PSII photochemistry (Fv/Fm), (**b**) effective quantum yield of PSII photochemistry (ΦPSII), (**c**) quantum yield of regulated (ΦNPQ), (**d**) non-regulated energy dissipation (ΦNO), (**e**) non-photochemical quenching (NPQ), and (**f**) electron transport rate (ETR). The analyses were conducted with 55 μmol m^−2^ s^−1^ of blue (450 nm) actinic light. Each data point represents the average ± SD of six independent measurements (*n* = 6). Different letters (a, b) indicate significant differences between treatments at *p* = 0.05 with a Tukey’s HSD test.

**Figure 9 ijms-25-09298-f009:**
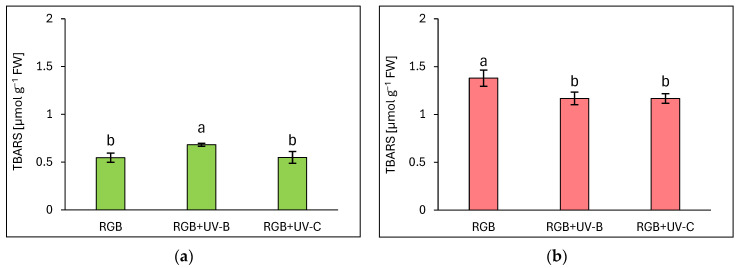
The rate of lipid peroxidation assessed with thiobarbituric acid reactive substances (TBARS) level of control (RGB), UV-B-treated (RGB + UV-B), or UV-C-treated (RGB + UV-C) plants of *baby leaf* lettuce (*Lactuca sativa* var. *crispa* L.) cultivars with (**a**) green (cv. Lollo Bionda) and (**b**) reddish leaf (cv. Lollo Rossa) at 20 DAS (days after sowing), estimated with TBARS assay. Each bar represents the average ± SD of six independent measurements (*n* = 6). Different letters (a, b) indicate significant differences between treatments at *p* = 0.05 with a Tukey’s HSD test.

**Figure 10 ijms-25-09298-f010:**
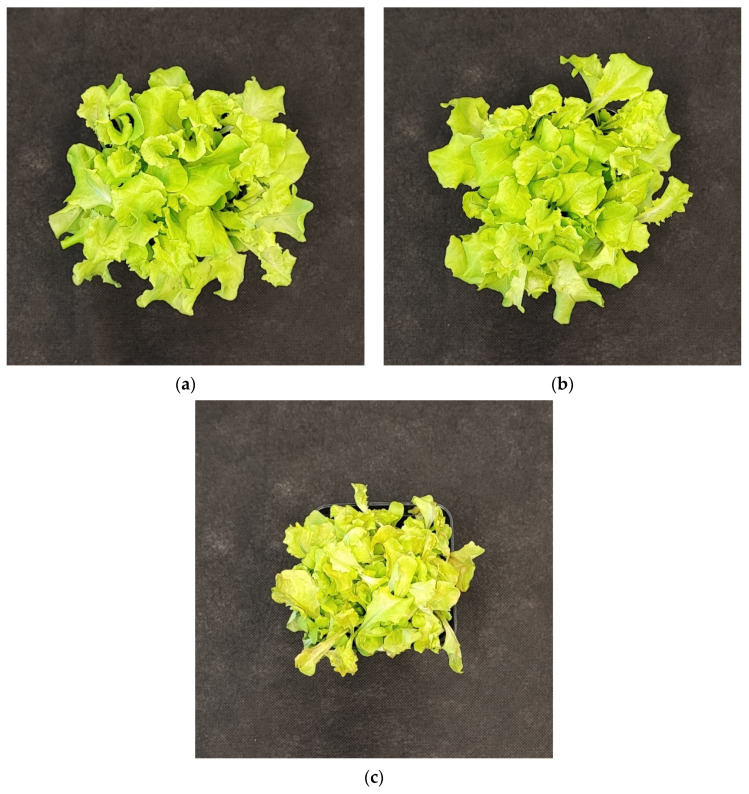
Morphology of 20-DAS plants of *baby leaf* lettuce (*Lactuca sativa* var. *crispa* L.) cultivar with green leaf (cv. Lollo Bionda, LB) grown under (**a**) RGB (C, control), (**b**) RGB + UV-B (UV-B supplemented, 311 nm), or (**c**) RGB + UV-C (UV-C supplemented, 254 nm) spectrum.

**Figure 11 ijms-25-09298-f011:**
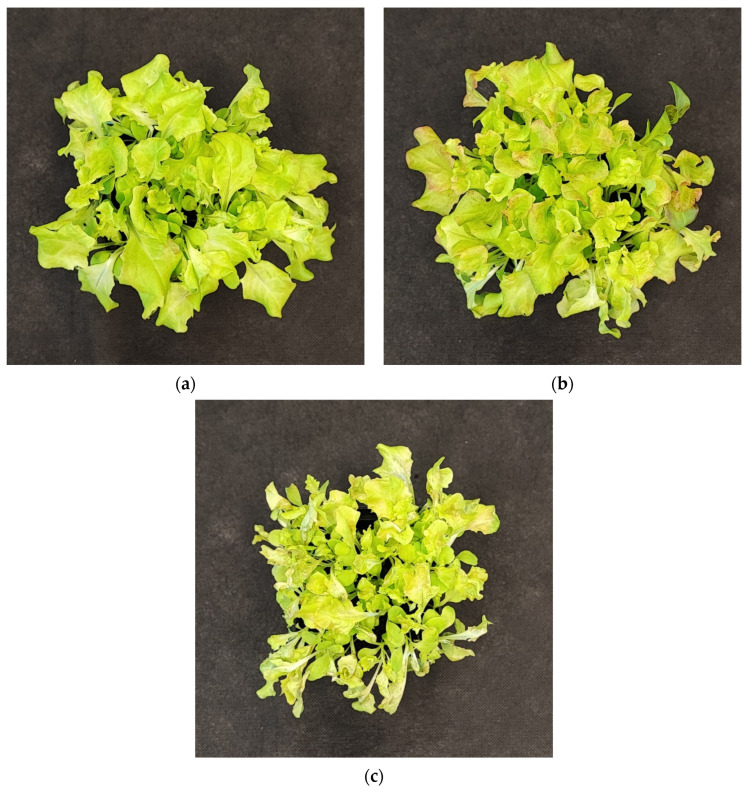
Morphology of 20-DAS plants of *baby leaf* lettuce (*Lactuca sativa* var. *crispa* L.) cultivar with reddish leaf (cv. Lollo Rossa, LR) grown under (**a**) RGB (C, control), (**b**) RGB + UV-B (UV-B supplemented, 311 nm), or (**c**) RGB + UV-C (UV-C supplemented, 254 nm) spectrum.

**Figure 12 ijms-25-09298-f012:**
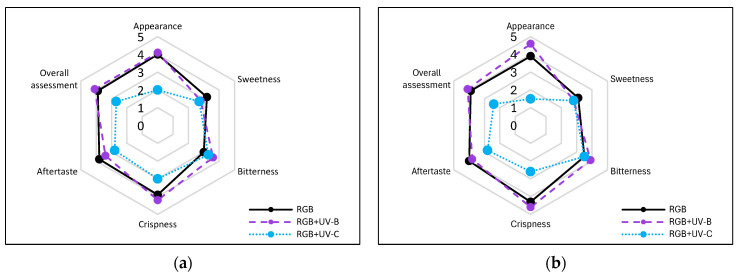
Sensory attributes of leaf samples of 20-DAS plants of *baby leaf* lettuce (*Lactuca sativa* var. *crispa* L.) cultivar with (**a**) green (cv. Lollo Bionda, LB) or (**b**) reddish leaf (cv. Lollo Bionda, LB) grown under RGB (C, control), RGB + UV-B (UV-B supplemented, 311 nm), or RGB + UV-C (UV-C supplemented, 254 nm) spectrum. The attributes of the blinded fresh leaf samples: appearance, sweetness, bitterness, crispness, aftertaste and overall assessment were scored at a scale of 0–5 (where 5 is maximal, 3—neutral, and 0—the least score) by 10 untrained consumers, aged 25–65 years (equal gender ratio). Each data point represents the average of ten independent tests (*n* = 10).

**Figure 13 ijms-25-09298-f013:**
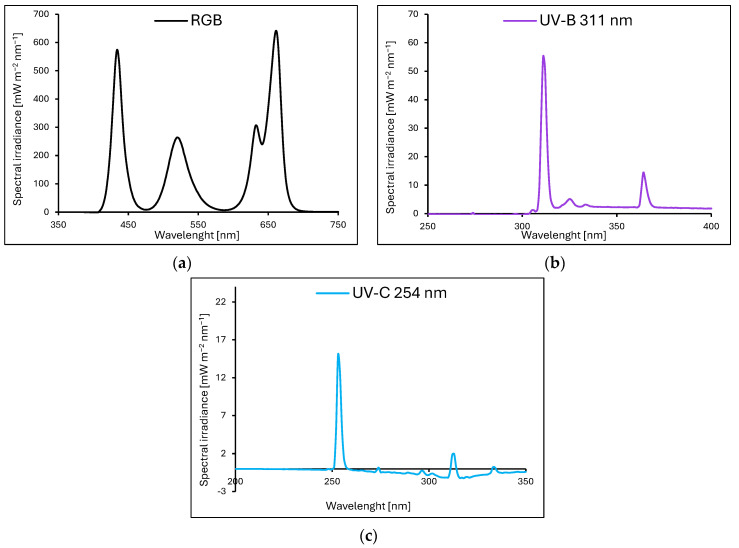
The light spectra of lamps were recorded with a spectroradiometer at four locations and then averaged. All plants tested were grown at 200 µmol m^−2^ s^−1^ of RGB (red–green–blue) spectrum (R–G–B; 661:633:520:434 nm) solely ((**a**), control) for 20 days or under RGB spectrum supplemented 4 days prior to harvest with increasing doses of UV-B (311 nm) (**b**) or UV-C (254 nm) (**c**).

**Table 1 ijms-25-09298-t001:** The abundance of photosynthetic pigments extracted with DMSO, the accumulation of soluble protein content (SLP) in leaves of *baby leaf* lettuce (*Lactuca sativa* var. *crispa* L.) cv. Lollo Bionda under different light conditions.

Parameter	Treatment		
	RGB	RGB + UV-B	RGB + UV-C
Chlorophyll *a + b* [mg g^−1^ FW]	0.839 ± 0.013 ^a^	0.764 ± 0.003 ^b^	0.323 ± 0.002 ^c^
Chlorophyll *a* [mg g^−1^ FW]	0.601 ± 0.010 ^a^	0.547 ± 0.002 ^b^	0.205 ± 0.001 ^c^
Chlorophyll *b* [mg g^−1^ FW]	0.238 ± 0.003 ^a^	0.217 ± 0.001 ^b^	0.117 ± 0.001 ^c^
Chlorophyll *a*/*b*	2.521 ± 0.016 ^a^	2.520 ± 0.002 ^a^	1.748 ± 0.010 ^b^
Carotenoids [mg g^−1^ FW]	0.114 ± 0.002 ^a^	0.112 ± 0.000 ^a^	0.031 ± 0.000 ^b^
Chlorophyll *a + b*/Carotenoids	7.373 ± 0.031 ^b^	6.807 ± 0.012 ^c^	10.361 ± 0.069 ^a^
Soluble leaf proteins [mg g^−1^ FW]	26.92 ± 0.15 ^c^	42.89 ± 1.12 ^a^	31.46 ± 0.57 ^b^

The presented values are means of six (or four for SLP) replicates ± SD. Different letters (a–c) in the same row indicate significant differences between treatments at *p* = 0.05 with a Tukey’s HSD test. FW—fresh weight.

**Table 2 ijms-25-09298-t002:** The abundance of photosynthetic pigments extracted with DMSO, the accumulation of soluble protein content (SLP) in leaves of *baby leaf* lettuce (*Lactuca sativa* var. *crispa* L.) cv. Lollo Rossa under different light conditions.

Parameter	Treatment		
	RGB	RGB + UV-B	RGB + UV-C
Chlorophyll *a + b* [mg g^−1^ FW]	0.824 ± 0.004 ^b^	0.836 ± 0.006 ^a^	0.798 ± 0.008 ^c^
Chlorophyll *a* [mg g^−1^ FW]	0.575 ± 0.003 ^a^	0.520 ± 0.003 ^c^	0.531 ± 0.004 ^b^
Chlorophyll *b* [mg g^−1^ FW]	0.250 ± 0.002 ^c^	0.316 ± 0.003 ^a^	0.267 ± 0.004 ^b^
Chlorophyll *a*/*b*	2.301 ± 0.011 ^a^	1.645 ± 0.011 ^c^	1.986 ± 0.018 ^b^
Carotenoids [mg g^−1^ FW]	0.124 ± 0.001 ^b^	0.141 ± 0.001 ^a^	0.092 ± 0.000 ^c^
Chlorophyll *a + b*/Carotenoids	6.645 ± 0.042 ^b^	5.936 ± 0.039 ^c^	8.651 ± 0.088 ^a^
Soluble leaf proteins [mg g^−1^ FW]	40.16 ± 0.15 ^a^	39.70 ± 1.20 ^a^	27.13 ± 1.72 ^b^

The presented values are means of six (or four for SLP) replicates ± SD. Different letters (a–c) in the same row indicate significant differences between treatments at *p* = 0.05 with a Tukey’s HSD test. FW—fresh weight.

**Table 3 ijms-25-09298-t003:** The schedule of supplemental UV-B and UV-C light treatment.

Treatment, Wavelength Peak (nm)	Daily Time Exposure (min), Diurnal Time				Total Time (h)	Total Irradiance (W m^−2^)	Irradiance (PAR) (W m^−2^)	Cumulative Dose (kJ m^−2^)
	Day 1	Day 2	Day 3	Day 4				
**UV-B, 311**	**15** **12.00–12.15 pm**	**30** **12.00–12.30 pm**	**60** **12.00–13.00 pm**	120 12.00–14.00 pm	3.75	1.1572	0.253	15.622
**UV-C, 254**	7.5 12.00–12.08 pm	15 12.00–12.15 pm	30 12.00–12.30 pm	60 12.00–13.00 pm	1.875	0.8901	0.177	6.008

UV-B states ultraviolet B light; UV-C states for ultraviolet C light; PAR states for photosynthetically active radiation.

## Data Availability

The data presented in this study are available on request from the corresponding author. The data are not publicly available due to the strict management of various data and technical resources within the research teams.

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
