# Peer review of "Effects of UV-B and UV-C Spectrum Supplementation on the Antioxidant Properties and Photosynthetic Activity of Lettuce Cultivars"

_ijms, 2024, doi:10.3390/ijms25179298_

Round 1

Reviewer 1 Report

Comments and Suggestions for Authors

The article “Effects of UV-B and UV-C spectrum supplementation on the antioxidant properties and photosynthetic activity of lettuce cultivars” by Skowron et al. discusses the changes in the content of polyphenolic compounds and antioxidant activity of two lettuce cultivars under the influence of stress factors such as ultraviolet radiation.

The introduction is sufficiently described. It introduces the subject of the article.

The methodology is well described. It contains all the important details allowing for a precise tracing of the stages of the conducted determinations.

The results and discussion are described with great accuracy.

Nevertheless, is it possible to standardize the unit for TPC and TFC? In one determination, the results were expressed in mmol, in the other in ug. I would also suggest assessing the profile of phenolic compounds using the UPLC method. This would show in detail the change in the content of individual phenols under the influence of the tested abiotic factors. The spectrophotometric methods selected in the article are not the most accurate.

It would also be worth considering conducting an organoleptic assessment. Did the use of UV-b and UV-C radiation, apart from visual changes in the product, also affect the taste of the raw material in any way?

The summary includes the most important elements of the manuscript. However, I miss a reference to the authors' future plans for the presented method or plans for its modification in order to increase the concentration of bioactive components.

Reviewer 2 Report

Comments and Suggestions for Authors

The manuscript demonstrates the potential of UV light in indoor farming to enhance lettuce's nutritional quality.

This article reveals the valuable application of UV light augmentation in indoor farming to improve lettuce's nutritional quality. UV-B radiation seems even more favorable, elevating antioxidant compounds without affecting the inhibition of photosynthesis than induction. Also, both green and red cultivars deranged in various physiological traits when the same cumulative dose of UV-C radiation was used. These findings offer useful knowledge on light quality management for indoor farming systems that harvest functional foods with high nutritional value.

I find this work very interesting, but I need to point out a few limitations, which authors can comment on in the manuscript. Firstly, the durations were short, hence the need to study both long-term and large samples. Then, this study was limited to two cultivars; therefore, future studies could analyze other varieties to generalize results. In addition, additional research is required to define the necessary dose and timing of UV supplementation. Moreover, this research scrutinizes the economic viability and energy use of large-scale indoor farming with supplementary UV. Commenting on this in the manuscript would make the article much better.

The abstract needs to be rewritten. It currently includes an introduction. Please focus on briefly describing what you did and achieved, without giving any details.

Please include Functional foods and Nutritional quality in the keywords. This will help your article reach a wider audience.

I have no concerns about the experimental section. The authors accurately described the methods and professionally presented the results.

I also have a question: How do the antioxidant levels in UV-treated lettuce compare to those grown under traditional outdoor conditions with natural sunlight exposure? Please include in the manuscript the comments.

Based on the above, I recommend that the article be published after minor revisions.
